# The cost- effectiveness of early dental visit in infants and toddlers focused on regional deprivation in South Korea: A retrospective cohort study

Eunsuk Ahn[1], Sun-Mi Kim[2]*

1 Department of Dental Hygiene, Daejeon Institute of Science and Technology, Daejeon, Republic of Korea,
2 Department of Dental Hygiene, Wonkwang Health Science University, North Jula, Republic of Korea

* dentalksm@naver.com

**Data Availability Statement:** Yes. doi: 10.6084/m9.figshare.14654442.

**Funding:** The authors received no specific funding for this work.

## Abstract

### Background

The aims of this study are to evaluate the cost-effectiveness of early dental visits (EDVs) and to investigate how regional deprivation impacts the economic evaluation.

### Methods

This study used the South Korea National Health Insurance database, which included medical claim data and voluntary-based oral examination data. The subjects of this study included whole participants for oral examinations for infants and toddlers of the National Health Insurance Corporation. A retrospective cohort study was designed and measured all oral treatments, costs, and number of visits for 208,969 children (experimental group, 101,768; non- experimental group, 107,201) who underwent oral examination for infants and toddlers from 2007 to 2014. The cost-effectiveness was measured using the incremental cost-effectiveness ratio, and the T-health index was used as the measurement for effectiveness. In addition, the difference in the effect according to the level of regional deprivation was confirmed.

### Results

The findings of this study showed that EDVs were cost-effective and that children who participated in EDVs had better oral health (T-health-2 index difference 0.32 point in most deprived regions) and needed 5 USD less costly dental treatments than those who did not have EDVs. The cost-effectiveness of EDVs varied according to the level of regional deprivation and was the highest in the most deprived regions.

### Conclusions

The study findings suggested that the provision of oral examination for infants and toddlers was a cost-effective dental policy. Additionally, EDVs were more effective in children who resided in the most deprived regions, a finding that will lead to the development of policy

**Competing interests:** The authors have declared that no competing interests exist.

intervention to improve dental care despite spatial inequality for disadvantaged population groups. Regarding the distribution of dental hospitals/clinics, incentive based dental polices for either dental providers or patients are needed that will assure the delivery of dental care despite spatial inequality.

## Introduction

Early childhood caries (ECC) is defined as at least one caries lesion in a child younger than 5 years of age [1, 2] and is recognized as a critical public health problem. Research findings show that the prevalence rate of ECC varies from population to population [2–5]. In the United States, the prevalence of dental caries among children 2–5 years of age was 27.9% between 1999 and 2004 [3]. It has been reported that 50% of Australian children under 12 years of age experience dental caries [4]. In Japan, the prevalence of dental caries was 25.9% among children 3 years of age [2]. The prevalence rate of disadvantaged groups in developing countries was as high as 85% [5]. In South Korea, the average prevalence rate of dental caries in 5-year-old children was 53.96% in 2007–2014 [6].

Previous studies reported that ECC not only affected oral health but also led to more pervasive consequences, such as costly treatment and issues with overall health, growth, and school performance [2, 7, 8]. ECC could negatively affect the quality of life for families [1, 2, 7] and could cause a significant economical and societal burden; however, it is preventable and potentially reversible in its early stages [9, 10]. With timely and early intervention, it is possible to prevent and eliminate future dental caries and thus reduce dental-related costs.

The guidelines for the prevention of ECC include the following: self-care, use of professional services, and exposure to community interventions such as water fluoridation [4, 9–11]. One of the prevention recommendations was the early dental visit (EDV), which was reported to reduce both dental caries and the burden of dental-related costs [5, 12]. The EDV was reported to be associated with fewer dental visits for children [5, 11]. Other studies found a relationship between the EDV and dental costs or treatment use [13, 14]. These studies emphasized that early prevention translated into a significant cost savings for dental care, especially for families at or below the poverty level. The research findings related to ECC led to the American Academy of Pediatric Dentistry, and the American Academy of Pediatrics recommendation [15] is that children have their first dental visit by their first birthday to prevent the incidence of ECC. In South Korea, the Oral Examination for Infants and Toddlers (which is the official name of the policy and database, OEIT) has been implemented as a part of the EDV since 2007. The OEIT is categorized into three age groups as follows: the 1st group is 18–24 months of age, the 2nd group is 42–48 months of age, and the 3rd group is 54–60 months of age.

Cost-effectiveness provides insight into the value of health interventions [16]. Findings of economic evaluations could help policy makers prioritize funding decisions. There are no empirical studies on the significant benefits and effectiveness of the oral examination for infants and toddlers in South Korea. Therefore, this study was conducted with the aim of confirming the effectiveness of the OEIT through the economic evaluation. In particular, it was attempted to determine the difference according to regional deprivation. And then, existing studies have used the dmft index, a conventional index for oral health, and have not confirmed oral health in terms of functionality, this study attempts to measure oral health using a new index that can measure functional oral health. The purpose of this study was twofold: (1) To

investigate whether children aged 5 years and younger who do not receive EDV have worse oral health than those who do receive EDV; and (2) To find if the cost-effectiveness differs according to the level of community deprivation and to determine if the most deprived areas are associated with greater effectiveness.

## Methods

### Data and subjects

This study used the database of the National Health Insurance (NHI), which included medical claim data, enrollee information of NHI, and voluntary-based OEIT data. The OEIT data is a complete data for examination of infants and children in South Korea, and the study was conducted without applying a sampling technique. South Korea attained universal health insurance in 1989. Since that time, all citizens have been entitled to the comprehensive NHI. This means that the NHI dataset of this study represents the total South Korean population [17]. Enrollee data included general information of NHI enrollees such as demographics, premiums, medical fees, and location of residences. The OEIT dataset encompassed information related to the oral health examination, such as caries and periodontal status, and the response to a questionnaire, which included behavioral information such as usage of oral hygiene devices, dietary habits, and nutrition. Since the OEIT is a voluntary-based national recipient program, the oral examination data only contained 32.75% of the total potential number of subjects.

### Study design

This study was a retrospective cohort study. The study population consisted of 208,696 recipients in the 3rd group OEIT, who were 54–60 months of age at the time of the oral examination in 2011–2014. We traced retrospectively for a maximum of 60 months and collected relevant information from all events (dental visits) including types of dental treatments, service costs, and number of visits (Fig 1) [6]. We operationally defined the EDV (exposed group) as the first dental visit in which dental services were utilized for oral examination prior to 24 months of age. In the non-EDV (non-exposed group), toddlers who did not undergo oral examination at 24 month were classified. A total of 208,969 were included as subjects. At this time, 101,768 people in the experimental group and 107,201 people in the non-experimental group were included, respectively. The study compared the cost-effectiveness between the 'exposed (EDV)' group and the 'non-exposed (non-EDV)' group.

This study was approved by the Wonkwang University Institutional Review Board (WKIRB-201602-SB-007).

### Data analysis

**Cost-effectiveness analysis.** The economic evaluation compares two or more health intervention programs through the examination of costs of inputs and health outcomes [18]. There are 3 types of economic evaluations as follows: cost-effectiveness analysis (CEA), cost-benefit analysis, and cost-utility analysis [18]. A cost-effectiveness analysis compares the costs with natural oral health outcome units such as the T-Health index [19], the DMFT (or dmft) index [9], and the number of missing teeth [20]. For the Tissue Health (T-health) index, the higher is given to the healthy tooth tissue, 4 points for healthy anterior teeth, 2 points for filling, 1 point for caries, and 0 points for loss [19]. A CEA is appropriate to inform decisions because it maintains health outcomes in its natural units rather than monetizing the outcomes [18, 21, 22]. Cost-effectiveness results can be expressed in costs per natural health units such as USD per

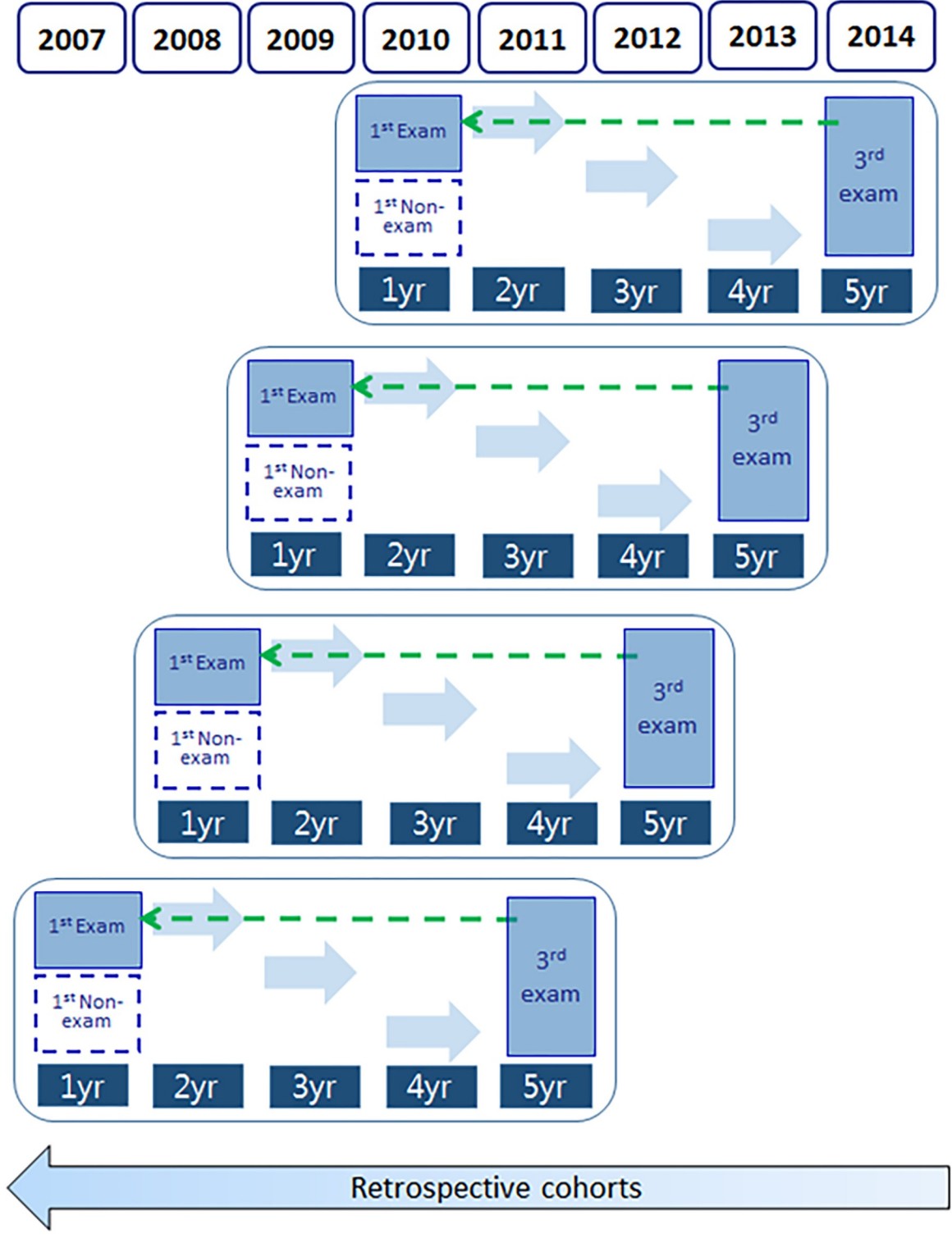

**Fig 1. Study population.**

the T-Health index or a cost-effectiveness ratio (CER) [16]. This study did not take into account the cost of loss of productivity and travel cost, so the cost-effectiveness of EDV and non-EDV compared by calculating the direct medical cost covered by insurance for five years after the infant oral examination, excluding the indirect cost. The cost-effectiveness analysis of this study was conducted from a community level, and the results were stated using a CER. All analyses were performed using STATA ver. 11.0 (Stata Corp., College Station, TX, USA).

**1) Costs.** Costs in the cost-effectiveness analysis are generally classified into direct costs and indirect costs. Indirect costs encompass loss of productivity and/or quality of life. In this study, direct costs were only considered because the study population included only children, and thus a loss in productivity was not relevant. The study also did not include the indirect costs of parents or guardians that accompanied the children because the dataset did not provide the information of the guardians and the length of time of stay for dental services. Currently, the cost of dental services is limited to the range covered by health insurance including the cost of dental services, such as examinations and treatments, and travel costs, which include the expense of transportation from the residence to the dental clinic.

**2) Effectiveness.** We defined effectiveness via the T-health index per child, which was introduced by Bernabé et al. [19]. The T-Health indicator is a weighted average of sound teeth, filled teeth, and teeth with some decay. The weights for T-Health represent the relative amounts of sound tissue in these three categories [23]. The T-Health index is based on the fact that a sound tooth contains more healthy dental tissue than a filled tooth and the latter can be considered to represent more healthy tissue than a decayed tooth because of the potential benefits of restorative treatment to the tooth shape and function [19, 23]. In this study, we used 5 measures of the T-Health index of Bernabé et al. [19]. In addition to the concept of the existing T-health index, an index with a weight change as shown in Fig 2 was used. This change in weight is because teeth with a smaller range of caries may have more healthy dental tissue than teeth with a wide range of filling. In this study, T-Health index is used to measure the magnitude of the effect. The T-Health index used in the study assigned a weight of 1 to a sound tooth, 0.10–0.50 to a filled tooth, 0.05–0.25 to a decayed tooth, and 0.00 to a missing tooth (Fig 2).

**3) Cost-effectiveness.** The dmft index was defined as a decayed teeth have untreated legion, filled teeth have been repaired with restorative treatment, and missing teeth have been extracted caused by dental caries. It was included for comparison with T-Health index.

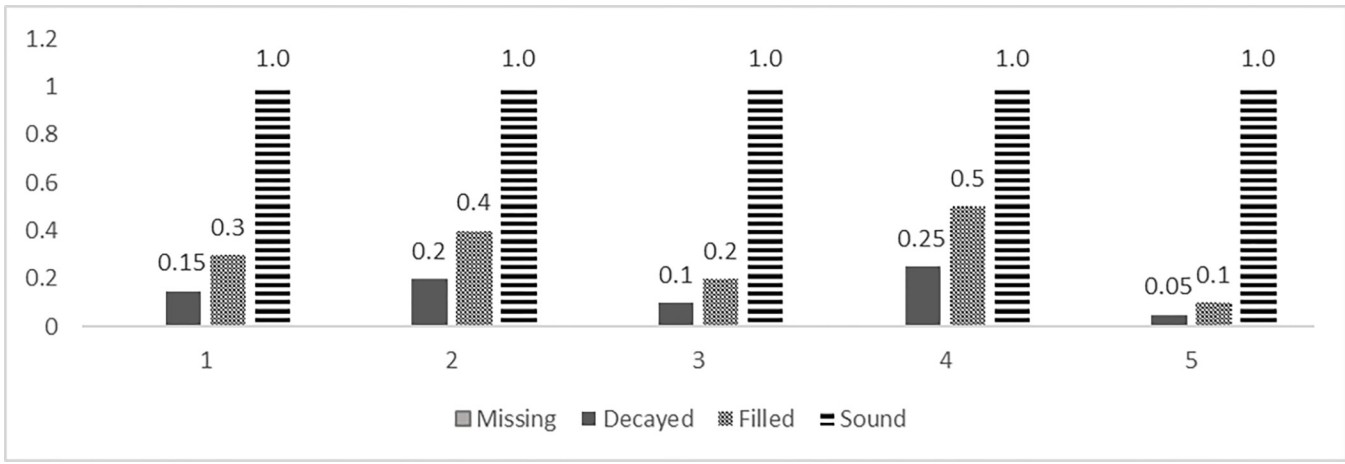

**Fig 2. Five measures of the T-Health Index\*.** Note: *This study used the T-Health index of Bernabé et al. [19].

In this study, effectiveness was measured using the incremental cost-effectiveness ratio, which was defined as follows:

$$ICER = \frac{cost_{EDV} - cost_{non-EDV}}{effect_{EDV} - effect_{non-EDV}}$$

where $cost_{EDV}$ and $cost_{non-EDV}$ per treated child represent the cost of the EDV group and the non-EDV group, respectively; $effect_{EDV}$ is the T-health index of the EDV group and $effect_{non-EDV}$ is the T-health index of the non-EDV group. The incremental cost-effectiveness ratio is a derivative concept and represents the incremental cost associated with additional unit of the T-health index.

**Considered variables.** We used gender, income, and regional variable for analysis. The income level was derived from the health insurance premium of the National Health Insurance program, because NHI is mandated for all citizens, and the premium is imposed on the basis of income or assets of the insured. A variable considered to the regional variable was the composite deprivation index (CDI). The CDI was composed of five different subdomains (unemployment, poverty, housing, labor, and social relations) and was used to indicate the socioeconomic status of a municipality [24]. The CDI score was categorized into 3 groups using tertile as follows: low, middle, and high, with the low group being considered more affluent.

## Results

### Study population distribution according to socio-demographic characteristics

Table 1 shows the general characteristics of the subjects. The overall EDV rate was 48.70% (101,768) and was higher in the affluent group. As the income level increased, the EDV rate tended to increase. The differences in the rates of EDV between most poor and most affluent income levels were large in the regions with a high CDI score, which is most deprived areas. The number of dental visits was higher in the EDV group and increased as the CDI score decreased. In contrast, the cost of dental services was inversely related to the number of visits. The cost of dental services was higher in the non-EDV group than in the EDV group (Table 1).

**Table 1. General characteristic of the subjects.**

| Variables | | Low CDI* (%) | | Middle CDI (%) | | High CDI (%) | | Total (person) |
|---|---|---|---|---|---|---|---|---|
| | | Non-EDV[a] | EDV | Non-EDV | EDV | Non-EDV | EDV | |
| Sex | Male | 33.23 | 17.03 | 34.29 | 16.10 | 34.15 | 16.09 | 105,108 |
| | Female | 32.69 | 17.04 | 33.60 | 16.01 | 33.80 | 15.96 | 103,861 |
| Income quintile | 1st quintile (most poor) | 9.71 | 4.53 | 11.81 | 4.96 | 13.22 | 5.37 | 34,517 |
| | 2nd quintile | 11.24 | 5.69 | 13.88 | 6.29 | 15.47 | 7.02 | 41,454 |
| | 3rd quintile | 8.00 | 4.30 | 9.10 | 4.51 | 9.54 | 4.66 | 27,923 |
| | 4th quintile | 12.53 | 6.90 | 12.80 | 6.64 | 12.32 | 6.40 | 40,115 |
| | 5th quintile (most affluent) | 24.45 | 12.64 | 20.31 | 9.71 | 17.40 | 8.60 | 64,960 |
| Average no. of visits | | 4.85 | 5.23 | 4.86 | 5.30 | 4.79 | 5.09 | |
| Cost of dental service[b] | | 226 | 221 | 234 | 232 | 246 | 235 | |

Asterisk (*) indicates that CDI: Composite deprivation index.

[a] EDV-early dental visit.

[b] unit-USD.

**Table 2. Effectiveness according to composite deprivation index.**

| | Low CDI* | | Middle CDI | | High CDI | |
|---|---|---|---|---|---|---|
| | Non-EDVᵃ | EDV | Non-EDV | EDV | Non-EDV | EDV |
| T-Health-10 | 17.29 | 17.52 | 17.31 | 17.50 | 17.27 | 17.56 |
| T-Health-14 | 17.46 | 17.68 | 17.48 | 17.67 | 17.45 | 17.72 |
| T-Health-6 | 17.12 | 17.36 | 17.13 | 17.34 | 17.09 | 17.39 |
| T-Health-18 | 17.63 | 17.84 | 17.66 | 17.84 | 17.63 | 17.89 |
| T-Health-2 | 16.95 | 17.20 | 16.96 | 17.17 | 16.91 | 17.23 |
| dmft index | 0.97 | 1.00 | 1.00 | 1.02 | 1.03 | 1.03 |

ᵃ EDV-early dental visit.

Asterisk (*) indicates that CDI: Composite deprivation index.

## The effectiveness of early dental visits

When comparing the T-Health index scores, the T-Health index scores of the EDV group were higher than those of its counterpart. The lowest scores occurred in the "middle CDI" regions, followed by the "low CDI" regions and the "high CDI" regions. Considering T-Health-2 as an example, the difference between the two groups (EDV vs. Non-EDV) in the "low" CDI regions was 0.25; the difference in the "middle" regions was 0.21; and the difference in the "high" regions was 0.32. However, the dmft index increased as the CDI score increased. The difference between the EDV group and the non-EDV group could be confirmed according to the type of T-health index whose measurement value was changed depending on whether or not dental treatment was received. However, the difference in the dmft index between the EDV and non-EDV groups was not consistent (Table 2).

## The cost-effectiveness of early dental visits

The ECC prevention effects of the EDV according to the regional deprivation index are shown in Fig 3. The incremental cost-effectiveness ratio of the T-health index using different weights showed cost-effectiveness for all weights.

At this time, negative ICER values that are shown in Fig 3 were due to positive effectiveness of natural units on teeth and the occurrence of treatment costs according to the condition of the teeth, which means a reduced cost between the non-EDV and EDV groups (Fig 3). The

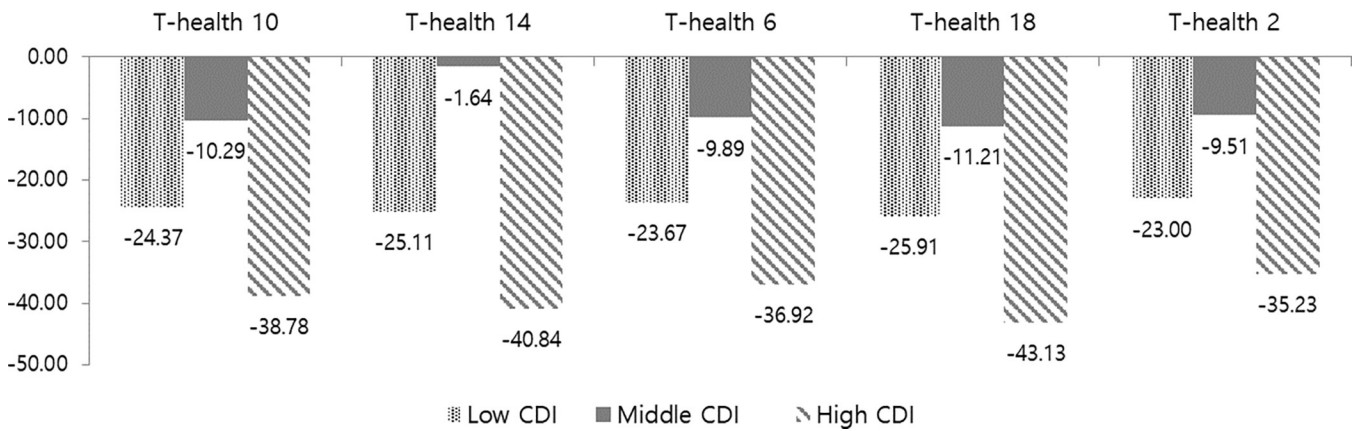

**Fig 3. ICER of early dental visits according to regional deprivation index.**

ICER varied according to the level of the CDI and was the highest in the high CDI regions. The ICER of the T-health index was distributed from 22.7 USD to 25.5 USD in the affluent regions. Among the middle CDI regions, the ICER of the T-health index showed from 1.6 USD to 10.9 USD, and in the regions with high CDI scores, the ICER was 34.5 USD to 42.7 USD. For the T-Health-10 index, the ICER of early dental visits in most deprived areas was approximately 39.1 USD, which indicated relatively high cost-effectiveness compared to 24.5 USD in the affluent areas.

## Discussion

Municipality socio-economic status could influence the oral health and accessibility of dental services, we also investigated how regional deprivation impacted incremental cost-effectiveness of EDVs. The results of this study showed that EDVs were cost-effective and that the effectiveness was the highest in the high CDI regions. In other words, these results show difference in dental accessibility by region and also in terms of cost-effectiveness. This means that preventive policies such as oral examinations for infants and toddlers can contribute to oral health equity [5].

In this study, comparing the average number of dental visits per child between the non-EDV group and the EDV group, as shown in Table 1, it was found that the average number of dental visits in the EDV group was higher than that of the non-EDV group. However, the opposite results were found when comparing the total cost of dental service. The difference in total cost of dental service between the two groups tended to increase in areas where regional deprivation was severe, which was a result of supporting the previous researches [10, 12].

The T-Health index was used instead of the dmft index to show the effectiveness of the ICER for EDVs. Many studies that involved economic evaluations used dmft/s, DMFT/s, or the potential cost savings associated with oral health outcomes [7, 11, 15, 18, 22]. However, the dmft/s or DMFT/s measures were less sensitive for detecting the progression of caries [25]. In addition, the functional oral health status was not well-reflected by the dmft index measures. Therefore, this study tried to improve these limitations by using the T-health index. As shown in Table 2, when the dmft index was used to evaluate the ICER, the effectiveness of the dmft index was higher in the EDV group, which is opposite to the expectation, because EDV users had more visits to the dental clinics for disease prevention. As a result, EDV users had several mild dental treatments, which resulted in a high score on the dmft. The dmft index does not also determine a change (better or worse) in oral health due to treatment for oral disease. However, the T-Health index score reflects the changes in the quality of diseased teeth [26]. As observed in this study, a higher T-health index score indicates a healthier oral status, and using the T-Health index helped to conclude that better oral health was maintained through early dental visits.

In this study, we found that the EDV group incurred less dental costs than the non-EDV group. In spite of the fact that the number of dental visits for children who experienced EDVs was higher than for children who did not participate in EDVs, the average cost per dental visit showed that children who experienced EDVs spent less. It was thought as a result of the timely intervention of dental services, which reduce the overall costs associated with dental treatments, and that untreated dental diseases became more severe and costly because treatments were postponed [1, 11, 13–15, 18]. The difference in dental costs between the two groups was the highest in the most-deprived regions. Previous studies reported that the behavior of individuals and available community resources were major determinants of oral health status [27–29]. It was assumed that children residing in deprived areas had inadequate access to dental services and, consequently, the oral health status of individuals was worse. When citing figures

of children 5 years of age from Statistics Korea in 2010 [30], the cost savings of EDVs was estimated at 3.6 million USD. The cost savings of this study might be underestimated, because this study only included minimal direct costs.

The proportion of total oral health care expenditure among gross domestic products (GDP) was 0.5% in Korea in 2019, overtaking 0.3% in the UK, similar to those in Japan and France. [31], which created a great economic burden on society. The economic evaluation of oral health interventions would help justify the priority of oral health programs based on a coherent scientific method [22, 25]. This study found that the cost-effectiveness differed according to the deprivation level of the community. The cost-effectiveness of the ICER in the most deprived areas compared to affluent areas was approximately 1.6 times higher on average. The results of this study are even more meaningful when consider that the burden of oral diseases is not only limited to economic burdens, but also includes pain, loss of productivity, and diminished quality of life. Since many researches are reported that children who come from a low socioeconomic status and disadvantaged community are at a relatively high risk for dental caries [28, 29, 32], providing timely oral services to children living in the most deprived communities result in greater cost savings and oral health improvements.

In conclusion, in terms of early intervention in the life course through early dental visits, providing oral examinations for infants and toddlers was a cost-effective policy. Additionally, EDVs were more effective in individuals who resided in the most deprived regions. This study is of the significance in that it suggested the necessity of a regional approach to the distribution of dental care by measuring the cost-effectiveness of EDV according to the level of regional deficiency using retrospective cohort data representing the Korean population. However, due to the limitation of the secondary data, there is a limitation in that indirect costs (travel cost, care cost etc), were not considered when calculating cost-effectiveness. Accordingly, in continued research, efforts to include appropriate variables that can sufficiently consider cost-effectiveness should be continued.

## Supporting information

**S1 File.**
(CSV)

**S2 File.**
(CSV)

**S3 File.**
(CSV)

**S4 File.**
(CSV)

**S5 File.**
(CSV)

## Author Contributions

**Conceptualization:** Eunsuk Ahn, Sun-Mi Kim.

**Data curation:** Eunsuk Ahn.

**Formal analysis:** Eunsuk Ahn.

**Investigation:** Eunsuk Ahn, Sun-Mi Kim.

**Methodology:** Eunsuk Ahn.

**Resources:** Eunsuk Ahn.

**Software:** Eunsuk Ahn, Sun-Mi Kim.

**Validation:** Eunsuk Ahn.

**Visualization:** Sun-Mi Kim.

**Writing – original draft:** Eunsuk Ahn.

**Writing – review & editing:** Sun-Mi Kim.

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
