## [Decision Letter · Decision Letter 0]

8 Apr 2021

PONE-D-20-34952

The cost-effectiveness of oral examination for infants and toddlers

PLOS ONE

Dear Dr. Kim,

Thank you for submitting your manuscript to PLOS ONE. After careful consideration, we feel that it has merit but does not fully meet PLOS ONE’s publication criteria as it currently stands. Therefore, we invite you to submit a revised version of the manuscript that addresses the points raised during the review process.

The major changes required by the reviewers is to strengthen the methodology, clarify some of the variables, define costs variables clearly and present the results for costs for defined variables for different regions for reference and control group separately. 

The result section has some missing information and discussion does not follow clearly from the results, 

Overall revision of manuscript addressing the reviewers comments is required. 

We look forward to receiving your revised manuscript.

Kind regards,

Charu C Garg, Ph.D.

Academic Editor

PLOS ONE

Journal Requirements:

Please include additional information regarding the data extraction tool used in the study and ensure that you have provided sufficient details that others could replicate the analyses. For instance, if you developed a data extraction tool as part of this study and it is not under a copyright more restrictive than CC-BY, please include a copy, in both the original language and English, as Supporting Information, or include a citation if it has been published previously.

In your Data Availability statement, you have not specified where the minimal data set underlying the results described in your manuscript can be found. PLOS defines a study's minimal data set as the underlying data used to reach the conclusions drawn in the manuscript and any additional data required to replicate the reported study findings in their entirety. All PLOS journals require that the minimal data set be made fully available. For more information about our data policy, please see http://journals.plos.org/plosone/s/data-availability.

Please include captions for your Supporting Information files at the end of your manuscript, and update any in-text citations to match accordingly. Please see our Supporting Information guidelines for more information: http://journals.plos.org/plosone/s/supporting-information.

Additional Editor Comments:

While the topic is important and interesting, the paper needs some revisions as mentioned by the two reviewers. The reviewers thought that the paper lacks methodological rigor in terms conclusions supporting the findings.

Additionally, some more comments are presented below

Title: There are two major aims of the paper, whereas the title does not refer to the community or regional deprivation at all. The long title could have the name of the country also.

Abstract : needs to be structured and some conclusions are presented even before the results eg. lines 33-35. Conclusions need to follow from the findings. Present the main cost findings in abstract.

Introduction: Please mention in introduction what we know from cost effectiveness studies in other countries and what is the value add for your study or is it for generating evidence in S Korea for what is known for other countries.

Methods: Please explain more clearly the different T-Health index used to measure outcomes for dental health

Line 120: reference is also made of DMFT index, but it does not have  a reference . Also mention how these indices are used to evaluate effectiveness in the paper.

Lines 124-125: Please explain what you mean by social perspective. In the next section, it is mentioned that the indirect and opportunity costs for households was not calculated.

Lines 134-136: please provide the detailed cost items included under the costs borne by health insurance. Are travel costs borne by households?

Line 147 – values are overlapping. Can you explain why and what that means. What are the 5 T-health indices imply in terms of effectiveness – especially when comparing the EDV ad NON-EDV group.

Line 160: what is the cost of the exposed group. Is it the cost per visit, cost per treated child or total costs of all children under the group? Clearly defines these groups.

Line 164: should it be average cost or incremental cost.

Results: clarify most poor and most affluent population – is it bottom 10% or 20% or 25%. Similarly, what top % it is? – table 1 shows how the population is divided for analysis- not clear why it is not 100% for either the EDV or the non-EDV group under any CDI score district.

Also not clear what is the n from which the % are calculated in table 1. Please give clearly the numbers under each group and for each variable considered.

Can you please explain the interpretation of values in table 2 with respect to different T-Health indices?

Before the results for ICER, the results should present a section on costs separately for EDV and Non EDV for different CDI regions. You mention societal costs in methods. What are the costs to the society for treating children with EDV and NON-EDV.

Discussion: Line 230: it should be cost effectiveness.

The discussion does not follow from results. Present the main results and then discuss in context of how these findings collaborate with what is known from literature.

Lines 233-255 – should follow from results. Costs are not mentioned separately for EDV and Non-EDV per dental visit for child. Will be useful to get the costs per treated child which would take into account higher no. of visits for children under EDV.

The discussion does not clearly specify dental costs of what is being considered in different places – eg. Line 261.

Line 267 – OECD data for 2013 is quoted. OECD at a glance will have much later figures. Otherwise also in introduction and discussion, several old references are used. Please try to update those.

Add strengths and limitations of the study. Also have the conclusion follow the results and discussion and not a generic conclusion.

Some of the variables mentioned in the supplementary information in in Korean. Please provide English translation.

Reviewers' comments:

Reviewer's Responses to Questions

**Comments to the Author**

1. Is the manuscript technically sound, and do the data support the conclusions?

Reviewer #1: Partly

Reviewer #2: Partly

2. Has the statistical analysis been performed appropriately and rigorously? 

Reviewer #1: No

Reviewer #2: No

3. Have the authors made all data underlying the findings in their manuscript fully available?

Reviewer #1: No

Reviewer #2: Yes

4. Is the manuscript presented in an intelligible fashion and written in standard English?

Reviewer #1: Yes

Reviewer #2: Yes

5. Review Comments to the Author

Reviewer #1: Comment: Kasahun Girma Tareke

The authors conducted a research entitled “The cost-effectiveness of oral examination for infants and toddlers”. The research is a very interesting work. However, there is a need to make amendment of important parts before accepted for publication. Here find the suggested comments to improve the paper.

Abstract

1. Please make partition for background, methods, results and conclusions.

1.1. Make your methods clear (i.e., sample size for each cases and controls, how you calculated the sample size, sampling technique, methods of data extraction from the data base, data analysis, etc.)

1.2. Make your results clear based on your research objective (s)/question (s). The result section seems like conclusions. Put the numeric/figurative findings for each of the findings.

1.3. Make your conclusions consistent with results. This might be corrected or becomes clear once you make correction of presentation of research findings.

Introduction

1. Good! However, make your research aims consistent with your title; given that the title only focused on cost-effectiveness but not included the first purpose of your study.

Materials and method

1. Please switch to “Methods”.

2. Please give descriptions of your study area/setting, and also the description of the project being implemented since 2007. You have mentioned that “The OEIT is categorized into three age groups 75 as follows: the 1st group is 18-24 months of age, the 2nd group is 42-48 months of age, and the 76 3rd group is 54-60 months of age.” Please specify the type of interventions done for each group.

3. Study design: It is good that you mentioned the study design. However, define the cases and controls clearly.

4. Study populations: It is not clear the number of population among cases and controls. Therefore, please specify it.

5. Sampling and sample size calculation: Your study lacks any information about the samples, calculation formulas, sampling techniques, procedures, etc.

6. Eligibility criteria??? Think of secondary data source.

7. Data analysis: It is not clear about the software you were used to clean, enter or analyzed the data. Other, the type of cost-analysis you have done is not consistent with your title. Your title only focused on one of the cost-analysis; cost-effective analysis.

8. Please make a measurement for your variables (cost-effectiveness, cost-benefit, and cost utility, cost, effectiveness, etc.)

9. Result: Good but you had not presented findings for the research purpose 1 presented on introduction section. Clearly show it. Also not presented according to cost-effectiveness, cost-benefit and cost-utility.

10. Discussion: Please delete the first two sentences of discussion and incorporate it somewhere in the introduction section. Write your pertinent findings as a first paragraph of discussion, and discuss them each by each in the consecutive paragraphs. The third paragraph seems like conclusion. What was your base to say cost-effective at this rudimentary stage? Make your study findings and discussion consistent. On line 271 of your paper, you have discussed that the cost-effectiveness of the ICER in the most deprived areas compared to affluent areas was approximately 1.6 times higher on 272 averages. But, you had not put the odds ratio for your findings. Therefore, please amend your result section incorporating the odds ratio, P-value and confidence intervals.

11. Please add strength and limitations of the study. I think it might have limitations since you had used a secondary data or retrospective cohort study.

12. Conclusions: Make a label “conclusions”. Make consistent with your objectives, pertinent findings and discussions.

13. Please define the abbreviations

14. Declerations????

15. References: You had used too old references. Please try to discuss your findings with the updated research findings.

Reviewer #2: Paper is well written but lack methodological rigour. It is not clear to me in what perspective health economic evaluation is done. Author mentioned social perspective, but in paper it does not reflect in terms of loss of productivity of parents who care for their children in lieu of using dental services. The ICER is not presented well. I can't make out any thing from graphs representing negative ICER bars. The paper clearly lack focus in terms of analysis, costs and benefits. The authors need to present cost-effectiveness overall and then look by level of deprivation. The deprivation cut-off levels are arbitrary and to nullify it it should be use tertile or quartie or quintile break-up.

6. PLOS authors have the option to publish the peer review history of their article (what does this mean?). If published, this will include your full peer review and any attached files.

Reviewer #1: **Yes: **Kasahun Girma Tareke

Reviewer #2: **Yes: **Anil Gumber

---

## [Author Response · Author response to Decision Letter 0]

11 Aug 2021

Reviewer 1. I have incorporated all of your suggestions into my revision. they were very helpful. Thank you. 

Reviewer 2. I have incorporated all of your suggestions into my revision. they were very helpful. Thank you.

---

## [Decision Letter · Decision Letter 1]

4 Dec 2021

PONE-D-20-34952R1The cost- effectiveness of early dental visit in infants and toddlers focused on regional deprivation in Korea: A retrospective cohort studyPLOS ONE

Dear Dr. Kim,

Thank you for submitting your manuscript to PLOS ONE. After careful consideration, we feel that it has merit but does not fully meet PLOS ONE’s publication criteria as it currently stands. Therefore, we invite you to submit a revised version of the manuscript that addresses the points raised during the review process. While the paper has been revised and submitted again as a fresh paper,  a rebuttal letter with specific answers to the reviewers comment's and how they were taken into account should have been provided. In fact the second reviewer still has several additional methodological comments and suggestions and would not like to accept the paper in current form.  we request the authors to take those into account and provide specific answers to each of the reviewers comment, as to how those were addressed.Please also provide answers to the comments that were given in version 1 of the comments.

We look forward to receiving your revised manuscript.

Kind regards,

Charu C Garg, Ph.D.

Academic Editor

PLOS ONE

Journal Requirements:

Additional Editor Comments (if provided):

Line 52: US data reported is quiter old. Can it be updated?

line 83: PLease the sentence construction. Seems like some missing information.

Lines 136 and 145 are repeated…. can be avoided

Tsble 1 - check quintiles

DMFT index values have not been explained in the methods, so not clear why it is contradictory to expectations in Line 269.

THe third effectiveness variable  - missing teeth  - has not been presented in the results.If not to be used, must be mentioned upfront in the methods.

please read thoroughly to correct the grammar. Several sentences do not read well.

Reviewers' comments:

Reviewer's Responses to Questions

**Comments to the Author**

1. If the authors have adequately addressed your comments raised in a previous round of review and you feel that this manuscript is now acceptable for publication, you may indicate that here to bypass the “Comments to the Author” section, enter your conflict of interest statement in the “Confidential to Editor” section, and submit your "Accept" recommendation.

Reviewer #1: (No Response)

Reviewer #2: (No Response)

2. Is the manuscript technically sound, and do the data support the conclusions?

Reviewer #1: (No Response)

Reviewer #2: Partly

3. Has the statistical analysis been performed appropriately and rigorously? 

Reviewer #1: (No Response)

Reviewer #2: No

4. Have the authors made all data underlying the findings in their manuscript fully available?

Reviewer #1: (No Response)

Reviewer #2: No

5. Is the manuscript presented in an intelligible fashion and written in standard English?

Reviewer #1: (No Response)

Reviewer #2: Yes

6. Review Comments to the Author

Reviewer #1: (No Response)

Reviewer #2: see my comments to the editor as authors have not fully addressed reviewers comments. They have not prepared response to reviewers.

7. PLOS authors have the option to publish the peer review history of their article (what does this mean?). If published, this will include your full peer review and any attached files.

Reviewer #1: **Yes: **Kasahun Girma Tareke

Reviewer #2: **Yes: **Anil Gumber

---

## [Author Response · Author response to Decision Letter 1]

11 Feb 2022

Thank you for taking a close look.

I made a mistake that should not be overlooked. As you said, we have corrected the title of the online system.

"The cost- effectiveness of early dental visit in infant and toddlers focused on regional deprivation in South Korea: A retrospective cohort study"

---

## [Editor Report · Decision Letter 2]

5 May 2022

PONE-D-20-34952R2

The cost- effectiveness of early dental visit in infant and toddlers focused on regional deprivation in South Korea: A retrospective cohort study

PLOS ONE

Dear Dr. Kim,

Thank you for submitting your manuscript to PLOS ONE. After careful consideration, we feel that it has merit but does not fully meet PLOS ONE’s publication criteria as it currently stands. Therefore, we invite you to submit a revised version of the manuscript that addresses the points raised during the review process.

There are few language issues and minor changes in some places in the attached document to bring more clarity to the paper. Some of the earlier edits suggested in the tables have not been made. For example 5 classes have been made and the authors still refer them to quartiles instead of quintiles.

We look forward to receiving your revised manuscript.

Kind regards,

Charu C Garg, Ph.D.

Academic Editor

PLOS ONE

Journal Requirements:

Additional Editor Comments:

Thank you for making changes as per the comments by the reviewers. Please see the attached copy for some minor clarifications and edits required for the document.
---

## [Author Response · Author response to Decision Letter 2]

13 May 2022

We thank the reviewers for their thorough review.

We have reviewed and revised the entire sentence as well as the part of the review comments.

The authors would like to thank the editorial staff once again for their efforts.

In addition, first author’s affiliation has been changed and the corresponding information is added.

Ahn Eunsuk

Division of Climate Change and Health Protection. Korea Disease Control and Prevention Agency.

---

## [Editor Report · Decision Letter 3]

31 May 2022

The cost- effectiveness of early dental visit in infants and toddlers focused on regional deprivation in South Korea: A retrospective cohort study

PONE-D-20-34952R3

Dear Dr. Kim,

We’re pleased to inform you that your manuscript has been judged scientifically suitable for publication and will be formally accepted for publication once it meets all outstanding technical requirements.

Kind regards,

Charu C Garg, Ph.D.

Academic Editor

PLOS ONE

Additional Editor Comments (optional):

There a a few minor editorials still required eg. line 38 - a word - "weight" seem to be missing. The editorial team may please see for correctness of grammar. 
---

## [Editor Report · Acceptance letter]

3 Jun 2022

PONE-D-20-34952R3 

The cost- effectiveness of early dental visit in infants and toddlers focused on regional deprivation in South Korea: A retrospective cohort study 

Dear Dr. Kim:

I'm pleased to inform you that your manuscript has been deemed suitable for publication in PLOS ONE. Congratulations! Your manuscript is now with our production department. 

Kind regards, 

on behalf of

Dr. Charu C Garg 

Academic Editor

PLOS ONE